# Chemically triggered drug release from an antibody-drug conjugate leads to potent antitumour activity in mice

Raffaella Rossin[1], Ron M. Versteegen[2], Jeremy Wu[3], Alisher Khasanov[4], Hans J. Wessels[5], Erik J. Steenbergen[6], Wolter ten Hoeve[7], Henk M. Janssen[2], Arthur H.A.M. van Onzen[1], Peter J. Hudson[3] & Marc S. Robillard[1]

Current antibody-drug conjugates (ADCs) target internalising receptors on cancer cells leading to intracellular drug release. Typically, only a subset of patients with solid tumours has sufficient expression of such a receptor, while there are suitable non-internalising receptors and stroma targets. Here, we demonstrate potent therapy in murine tumour models using a non-internalising ADC that releases its drugs upon a click reaction with a chemical activator, which is administered in a second step. This was enabled by the development of a diabody-based ADC with a high tumour uptake and very low retention in healthy tissues, allowing systemic administration of the activator 2 days later, leading to efficient and selective activation throughout the tumour. In contrast, the analogous ADC comprising the protease-cleavable linker used in the FDA approved ADC Adcetris is not effective in these tumour models. This first-in-class ADC holds promise for a broader applicability of ADCs across patient populations.

[1] Tagworks Pharmaceuticals, Geert Grooteplein Zuid 10, 6525 GA Nijmegen, The Netherlands. [2] SyMO-Chem B.V., Den Dolech 2, 5612 AZ Eindhoven, The Netherlands. [3] Avipep Pty Ltd, 343 Royal Parade, Parkville, VIC 3052, Australia. [4] Levena Biopharma, 4955 Directors Place, Suite 300, San Diego, CA 92121, USA. [5] Radboud Proteomics Centre, Department of Laboratory Medicine, Radboud University Medical Center, P.O. Box 9101, 6500 HB Nijmegen, The Netherlands. [6] Department of Pathology, Radboud University Medical Center, P.O. Box 9101, 6500 HB Nijmegen, The Netherlands. [7] Syncom B.V., Kadijk 3, 9747 AT Groningen, The Netherlands. Correspondence and requests for materials should be addressed to M.S.R. (email: marc.robillard@tagworkspharma.com)

Antibody-drug conjugates (ADCs) are a promising class of biopharmaceuticals that combine the target-specificity of monoclonal antibodies (mAbs) or mAb fragments with the potency of small molecule toxins[1,2]. They are designed to bind to an internalising cancer cell receptor leading to uptake of the ADC and subsequent intracellular release of the drug by enzymes, thiols or lysosomal pH. Routing the toxin to the tumour, while minimising the peripheral damage to healthy tissue, allows the use of highly potent drugs resulting in improved therapeutic outcomes. Presently, four ADCs are approved by the American Food and Drug Administration (FDA): brentuximab vedotin (Adcetris) for Hodgkin and anaplastic large cell lymphoma, ado-trastuzumab emtansine (Kadcyla) for HER2-positive metastatic breast cancer, gemtuzumab ozogamicin (Mylotarg) for acute myeloid leukaemia and inotuzumab ozogamicin (Besponsa) for the treatment of acute lymphoblastic leukaemia. For example, Adcetris afforded a 75% overall response rate in patients with relapsed or refractory Hodgkin lymphoma and a median duration of response of 21 months[3]. Encouraged by these first successes, over 60 ADCs are now in clinical trials for a variety of haematologic and solid tumour malignancies[1–3].

Nevertheless, the current strategies do have some limitations, especially with respect to solid tumours. Haematologic tumours typically exhibit specific and homogenous expression of the target antigen and are well perfused and, therefore, accessible to the ADC[3,4]. On the other hand, therapy of solid tumours is hampered by the relatively limited number of suitable cancer-specific targets and the poor intratumoral distribution of ADCs[2,4]. The elevated interstitial pressure in solid tumours impedes penetration by large constructs such as ADCs[5]. This penetration can also be affected by the binding to cancer cells in the perivascular space and to antigens in the interstitial space, shed from dying cells[5–7]. The heterogeneous receptor expression observed in solid tumours further confounds homogenous drug delivery[1,5]. Importantly, the number of solid tumour-specific receptors that ensure efficient internalisation and drug release is relatively limited. Low

receptor copy numbers, slow internalisation kinetics, inefficient subcellular trafficking and receptor expression levels in normal tissues all complicate the selection of solid tumour targets for the current ADC approaches[4,8,9]. Furthermore, contrary to haematologic targets, solid tumour targets are typically only overexpressed in a subset of patient populations[3,8]. For example, only 20% of breast cancer patients have sufficient HER2 expression to be eligible for treatment with Kadcyla.

An approach that functions by means of extracellular drug release would expand the number of potential ADC targets as there are sufficient non-internalising receptors and extracellular matrix targets that are selectively present in solid tumours[10–15]. Such targets may become amendable to ADC therapy by using a bioorthogonal chemical reaction for selective antibody-drug cleavage in vivo instead of relying on intracellular biological activation mechanisms. In this two-step approach, tumour binding of the ADC and blood clearance of the unbound fraction would be followed by systemic administration of an activator that reacts with the ADC linker, leading to drug release and subsequent uptake into surrounding cancer cells as well as tumour-supporting stromal cells (Fig. 1). Indeed, extracellular cleavage of disulphide- and peptide-linked ADCs by endogenous mechanisms has recently been shown to afford therapeutic efficacy in several mouse models, while it is generally accepted that such linkers need to be internalised to achieve sufficient cleavage[16,17]. Likewise, a chemically cleavable ADC system would expand the target scope and, in contrast to the inherent variability that can hamper endogenous mechanisms, would enable universal and direct temporal control over drug release. Furthermore, extracellular release could possibly allow more drug to diffuse into the tumour, aiding homogenous drug delivery and the bystander effect, thus potentially improving therapeutic efficacy in heterogeneous or poorly penetrated tumours[1,2]. The fastest bioorthogonal (click) reaction, the inverse-electron-demand Diels−Alder (IEDDA) conjugation between trans-cyclooctene (TCO) and tetrazine derivatives, is now an established method for bioconjugations in mice[18]. We showed that this IEDDA conjugation

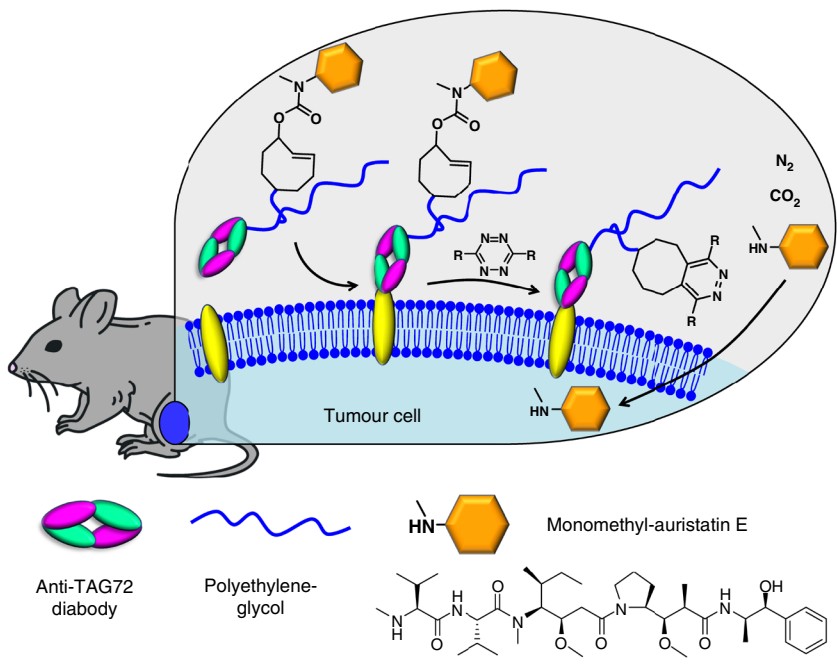

**Fig. 1** Triggered drug release using "click-to-release" chemistry in vivo: on-tumour liberation of a cell permeable drug (monomethyl auristatin E, MMAE) from a trans-cyclooctene-linked ADC following systemic administration of a tetrazine activator

Anti-TAG72 diabody

Polyethylene-glycol

Monomethyl-auristatin E

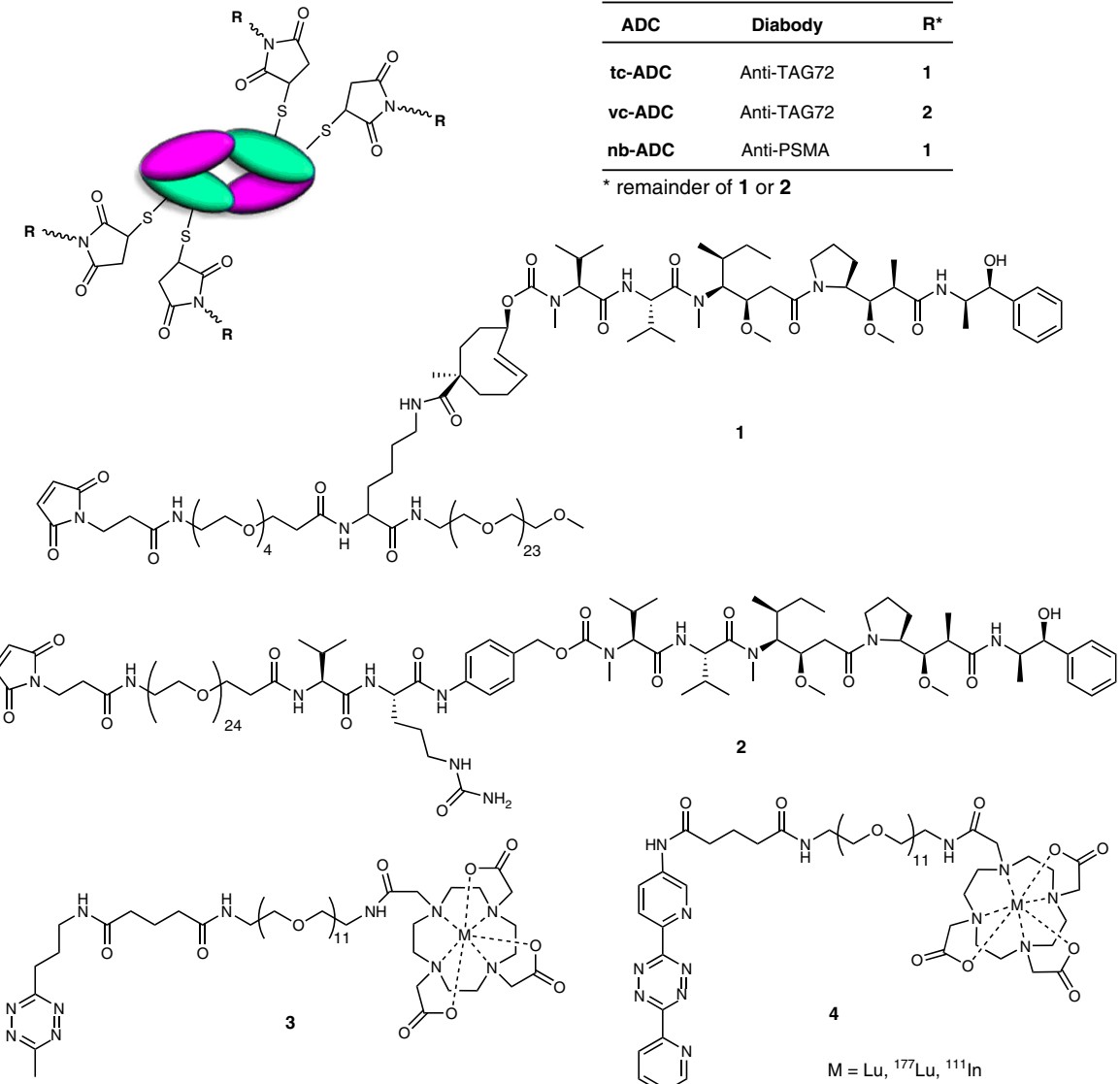

| ADC | Diabody | R* |
|---|---|---|
| **tc-ADC** | Anti-TAG72 | 1 |
| **vc-ADC** | Anti-TAG72 | 2 |
| **nb-ADC** | Anti-PSMA | 1 |

\* remainder of **1** or **2**

M = Lu, [177]Lu, [111]In

**Fig. 2** Compounds used in this study: diabody-based ADCs **tc-ADC** (anti-TAG72, containing the chemically cleavable TCO linker), **vc-ADC** (anti-TAG72, containing the enzymatically cleavable valine-citrulline linker), and **nb-ADC** (non-binding, anti-PSMA, containing the TCO linker), linker-drug building blocks (**1** and **2**), and tetrazine-containing activator (**3**) and probe (**4**)

could be transformed into an IEDDA pyridazine elimination reaction leading to the cleavage of allylic carbamates from TCO upon reaction with tetrazine[19], opening up bioorthogonal "click-to-release" applications in vitro and in vivo[18,20–24]. This reaction seemed well suited for the envisioned chemically triggered ADC therapy, which may address the relatively limited number of clinically validated solid tumour targets for the current systems.

Herein, we report the first example of such click-to-release ADC therapy in tumour-bearing mice, enabled by the development of a diabody conjugate with a high tumour uptake combined with a fast blood clearance, and a new tetrazine activator that gives near-quantitative drug release (Fig. 1). Biodistribution and imaging experiments show that the ADC, comprising the TCO-linked drug monomethyl auristatin E (MMAE) and polyethylene glycol (PEG), matches the high tumour uptake of the parent mAb combined with very low levels in non-target tissues, and that the tetrazine activator can effectively reach this conjugate throughout the tumour. Efficacy studies in two mouse xenograft models demonstrate a potent therapeutic effect, whereas an analogous ADC containing the protease

sensitive valine-citrulline linker, used in the marketed ADC Adcetris and designed for intracellular release, fails to control tumour growth in both models. These findings indicate that the click-to-release concept allows to expand the scope of ADC therapy to non-internalising receptors and stroma targets, forming the basis for a broader applicability across patient populations, thus potentially lowering the hurdle for success against solid tumours.

## Results

**Development of the anti-TAG72 ADC.** We chose to use the tubulin-binding antimitotic MMAE as the toxin as it is widely used in ADCs and, essential for our approach, is cell permeable[1]. Tumour-associated glycoprotein-72 (TAG72) was selected as the cancer target, as it does not internalise, has slow shedding, and is widely expressed in a range of epithelial-derived human adeno-carcinomas such as breast, colorectal, stomach, lung, pancreatic, prostate and ovarian cancer, while TAG72 expression in normal adult tissues is very low[15]. The TAG72 targeting mAb CC49 has

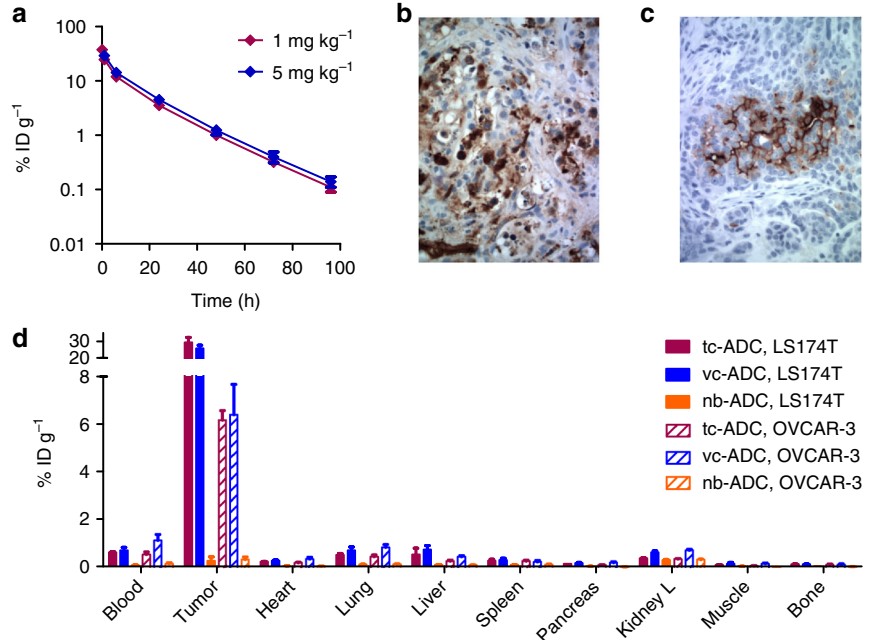

**Fig. 3** Biodistribution of diabody-based anti-TAG72 **tc-ADC** and the control ADCs **vc-ADC** (valine-citrulline linker; anti-TAG72) and **nb-ADC** (non-binding, anti-PSMA; TCO linker). **a** Blood kinetics of [125]I-labelled **tc-ADC** administered i.v. at 1 and 5 mg kg$^{-1}$ dose in tumour-free mice (calculated half-lives of 5.95 and 7.17 h, respectively). Immunostaining of **b** LS174T colon carcinoma and **c** OVCAR-3 ovarian carcinoma slices showing TAG72 expression (brown staining). **d** Biodistribution of [125]I-labelled **tc-ADC** and controls **vc-ADC** and **nb-ADC** (2 mg kg$^{-1}$) 48 h post-injection i.v. in mice-bearing subcutaneous LS174T and OVCAR-3 xenografts, respectively; blood level of **tc-ADC** comparable to **vc-ADC** and to level in tumour-free mice at 48 h (Fig. 3a), while **nb-ADC** levels in blood and other tissues are lower due to faster clearance. Data represent the mean percentage injected dose per gram (% ID g$^{-1}$) with s.d. ($n = 4$)

been used in the clinic for (pretargeted) radioimmunotherapy of ovarian, colorectal, lung, prostate and breast cancer[25,26] and for preclinical on-tumour IEDDA conjugation and elimination reactions[23,27,28]. Due to the slow clearance of mAbs from blood, a mAb-based ADC would require a very long interval between ADC and activator administration in order to prevent undesired drug release in circulation and well perfused tissues. To minimise this interval, we decided to develop a faster clearing PEGylated CC49 diabody conjugate instead. Conjugation of PEG$_{24}$ residues to a CC49 diabody had previously been shown to afford superior imaging properties compared to both the native diabody and the parent mAb, owing to a high tumour uptake combined with a complete blood clearance within 2–3 days[29]. In addition to the likelihood of very low off-target activation levels, a smaller ADC lacking an Fc region could also improve tumour penetration and preclude the toxicities from Fc cross reaction with normal tissues as seen for current ADCs[2,30]. In the context of pretargeted radioimmunoimaging it had been demonstrated that mAb-conjugated TCO can be isomerised to the unreactive cis-isomer through contact with copper-binding sites in the serum proteins albumin and ceruloplasmin and that this deactivation can be slowed down to half-lives of 5 days or more in mice by reducing the mAb-TCO spacer length, thereby shielding the TCO[31]. To achieve the same in the ADC, we prepared a drug linker comprising a lysine-branched spacer designed to have the TCO-MMAE shielded by the vicinity of the diabody as well as the PEG$_{24}$ moiety (**1**, Fig. 2). This maleimide-functionalized TCO-MMAE linker **1** was site-specifically conjugated to four engineered cysteine residues in the CC49 diabody providing **tc-ADC** (61 kDa, Fig. 2) with a drug-to-antibody ratio (DAR) of 4 and with complete retention of immunoreactivity towards its target TAG72 (Supplementary Fig. 9). The conjugate showed excellent stock stability (phosphate-buffered saline (PBS), 4 °C) as no TCO

isomerisation or spontaneous drug liberation was observed in 6 months (Supplementary Figs. 2a, 2b and 4). In addition, no drug release was observed upon incubation of the ADC in serum at 37 °C for 24 h (Supplementary Fig. 6).

Next, we examined the pharmacokinetics of intravenous (i.v.) administered [125]I-labelled **tc-ADC** in tumour-free mice (Fig. 3a, Supplementary Tables 3 and 4) as well as its biodistribution in two TAG72-expressing tumour models: mice-bearing colon carcinoma xenografts (LS174T) characterised by a relatively high TAG72 expression, and mice-bearing ovarian carcinoma xeno-grafts (OVCAR-3) with a lower TAG72 expression (Fig. 3b–d and Supplementary Fig. 13). The clearance was found to be nearly complete 2 days post-injection (Fig. 3a), at which point the tumour uptake in LS174T-tumored mice was 29% injected dose per gram (ID g$^{-1}$) (Fig. 3d), matching the high uptake of the native CC49 mAb in this animal model[27,28] and demonstrating that the diabody conjugate design combines a low systemic exposure with a high tumour accumulation as compared with other mAb fragment drug conjugates[30]. As expected, the uptake in the OVCAR-3 xenografts was lower at 6% ID g$^{-1}$, which is still much higher than the 0.8% ID g$^{-1}$ uptake in TAG72-negative HT-29-tumour-bearing mice (Supplementary Table 5). The very low retention of **tc-ADC** in blood (<1% ID g$^{-1}$) and especially in non-target tissues for both models indicated that an interval of 2 days between ADC and activator administration should lead to ADC activation with good tumour selectivity. To assess the serum protein-induced deactivation rate of the TCO linker in vivo, mice were administered [125]I-labelled **tc-ADC** and blood samples were obtained over time and reacted ex vivo with an excess of the highly reactive [177]Lu-labelled 3,6-bispyridyl-tetrazine-DOTA probe **4** (Fig. 2), resulting in IEDDA conjugation of **4** to the TCO linker of the ADC. Separation of the ADC from unreacted [[177]Lu]Lu-**4** and quantification of the [177]Lu/[125]I ratio led to a

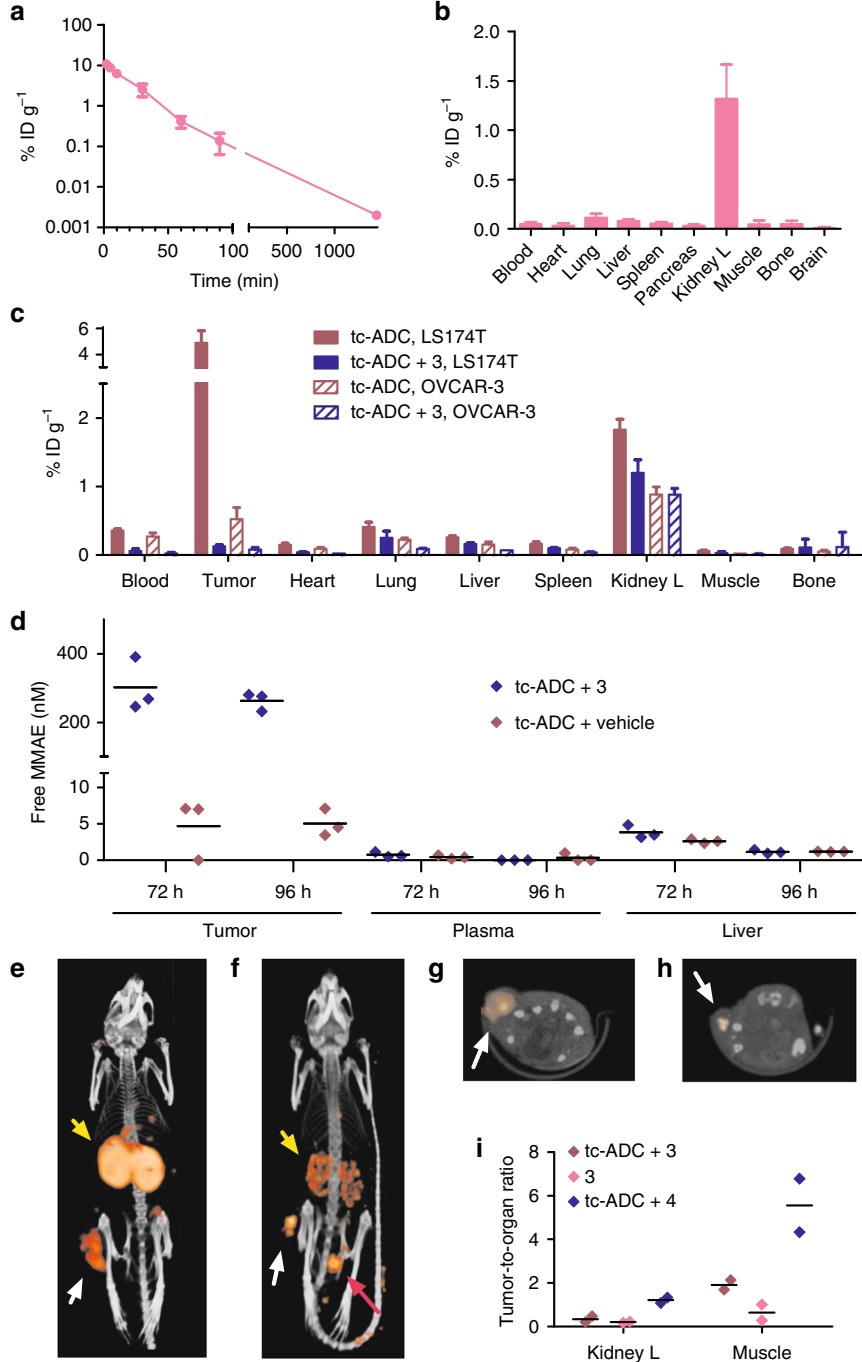

**Fig. 4** In vivo assessment of activator **3** and probe **4**. **a** Blood kinetics of [177]Lu-labelled **3** administered i.v. at 0.335 mmol kg[−1] (calculated half-life: 12 min) and **b** biodistribution of [[177]Lu]Lu-**3** in tumour-free mice 1 h post-injection. **c** Blocking study: biodistribution of probe [[177]Lu]Lu-**4** (0.335 μmol kg[−1]) 3 h post-injection i.v. in LS174T and OVCAR-3 tumour-bearing mice pretreated i.v. with **tc-ADC** only (2 mg kg[−1]) or with **tc-ADC** followed by **3** (0.335 mmol kg[−1]); data represent the mean percentage injected dose per gram (% ID g[−1]) with s.d. (n = 4). **d** MMAE concentration in LS174T xenografts, plasma and livers of mice injected with **tc-ADC** (2 mg kg[−1]; n = 3) followed 48 h later by activator **3** (0.335 mmol kg[−1]) or vehicle and euthanized 72 or 96 h post-ADC administration. **e** Post-mortem SPECT/CT projection of a LS174T tumour-bearing mouse pretreated i.v. with **tc-ADC** (2 mg kg[−1]) followed 48 h later by the [111]In-analogue of **3** (13 nmol kg[−1], 13 MBq, i.v.). **f** SPECT/CT projection of a live LS174T tumour-bearing mouse pretreated i.v. with **tc-ADC** (2 mg kg[−1]) followed 48 h later by [111]In-labelled **4** (19 nmol kg[−1], 24 MBq, i.v.). **g**, **h** Single transverse slice passing through the tumour of the mouse in resp. **e**, **f** White arrows indicate the tumour; yellow arrows indicate the kidneys; pink arrow indicates the bladder. **i** Tumour-to-organ ratios of [[111]In]In-**3** with and without pre-administration of **tc-ADC** and tumour-to-organ ratios of [[111]In]In-**4** with pre-administration of **tc-ADC**, derived from the SPECT/CT imaging (n = 2)

deactivation half-life of ca. 5.5 days (Supplementary Fig. 15), matching that of other TCOs[23,31] and exceeding the 2 days needed for the ADC to clear from blood, after which the tumour-bound fraction can be activated to induce drug release.

**Development of the tetrazine activator**. The characteristics of **tc-ADC** held promise for efficient tumour therapy provided that a tetrazine activator could be found that enables quantitative reaction with the tumour-bound ADC and exhibits a high release

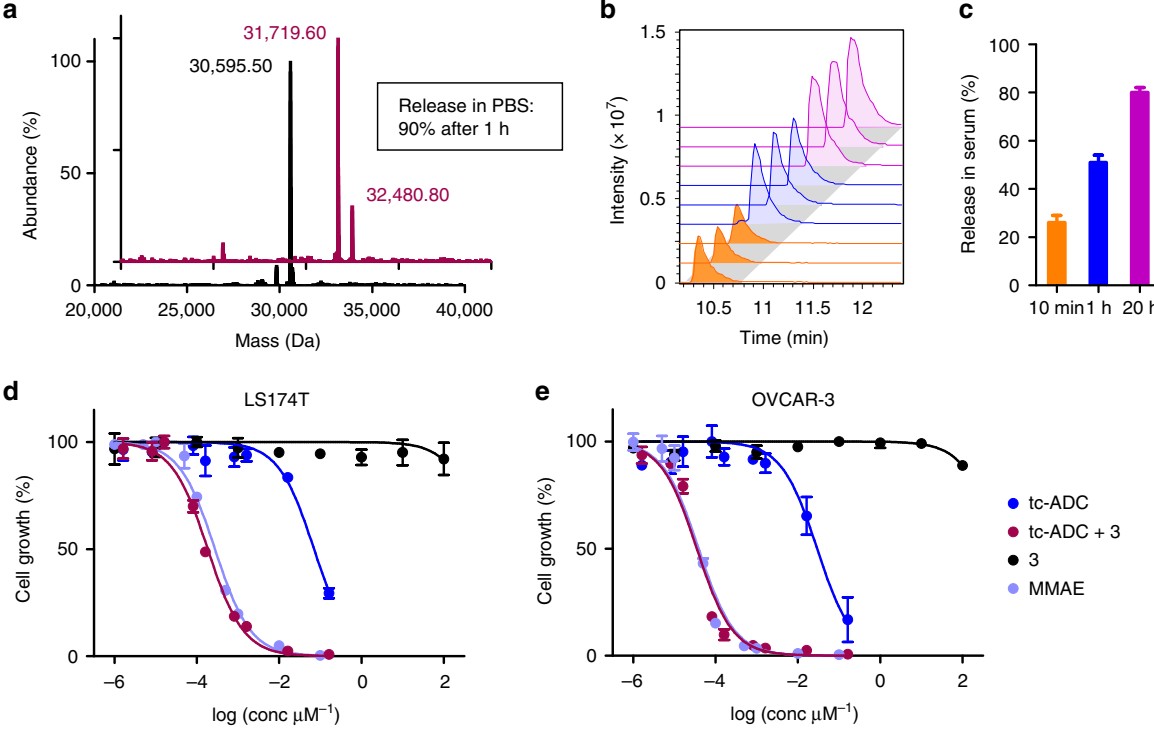

**Fig. 5** Tetrazine-triggered drug release from **tc-ADC** in vitro. **a** HPLC-QTOF-MS analysis of **tc-ADC** showing, after deconvolution, the fully MMAE-conjugated monomer (30,595.50 Da) and, after a 1 h reaction with **3** in PBS at 37 °C, the monomer with two pyridazine derivatives and no MMAE (31719.60 Da: 80%) and the monomer with two pyridazine derivatives and one MMAE still attached (32480.80 Da: 20%). **b** HPLC-QTOF-MS analysis of deproteinized mixtures of **tc-ADC** and **3** (in triplicate) after 10 min (orange), 1 h (blue) and 20 h (magenta) incubation in 50% serum at 37 °C; **c** quantification of recovered MMAE using calibration curves. Cytotoxicity of combined **tc-ADC** and **3** in **d** LS174T cells (EC$_{50}$ = 185 pM) and **e** OVCAR-3 cells (EC$_{50}$ = 35 pM) in vitro, in comparison with only **tc-ADC** or **3**; free MMAE (EC$_{50}$ = 277 and 39 pM) was used as positive control. As the ADC activation was not performed on the cell surface but in solution in the presence of the cells, similar results were obtained for the TAG72-negative HT-29 cells, see Supplementary Table 1 and Supplementary Fig. 12

yield. The 3,6-bisalkyl-tetrazine and 3-alkyl-6-aryl-tetrazine structures that afford cleavage have a lower reactivity compared to the highly reactive but poorly releasing 3,6-bispyridyl-tetrazine motif (present in **4**), which has successfully been used in IEDDA conjugations in vivo[18,19,28,32]. As small molecules, these less reactive tetrazines likely clear from circulation too fast, precluding quantitative on-tumour ADC reaction[23]. For example, small 3,6-bisalkyl-tetrazines have been shown to clear with half-lives of ca. 1 min[33,34]. Conjugation to 10 kDa dextran with a clearance half-life of ca. 6 min[35] partially remedied this but at the expense of a substantially lower maximal release yield of 50%, possibly due to the dextran microenvironment affecting the formation of the releasing intermediate[23]. We, therefore, aimed to replace the dextran by a smaller moiety that would slow down the blood clearance without negatively impacting the release yield provided by the 3,6-bisalkyl-tetrazine motif. Intrigued by the relatively slow clearance (10 min half-life) of the 3,6-bispyridyl-tetrazine imaging probe **4**, containing a tetraazacyclododecane-1,4,7,10-tetraacetic acid (DOTA) chelate, used in earlier extracellular pretargeted imaging studies[28], we set out to identify the governing factor. We prepared an analogue of **4** comprising a PEG$_{11}$-linked DOTA as well as an oxime linkage to an $^{18}$F-labelled fluorobenzaldehyde ([$^{18}$F]F-**S24**, Supplementary Fig. 14a). Pharmacokinetic and tumour pretargeting studies using pre-administered TCO-tagged CC49 demonstrated that the blood clearance and tumour uptake of this probe were similar to those of probe **4** (Supplementary Fig. 14 and Supplementary Table 2). Surprisingly, [$^{18}$F]fluorobenzaldehyde-labelled derivatives without the chelate, but with a PEG$_{11}$, a PEG$_{11}$-glutamic acid, or only a glutamic acid moiety all

showed a markedly faster blood clearance and a corresponding lower tumour capture by pre-administered TCO-tagged CC49 (Supplementary Fig. 14). This prompted us to exploit the unexpected clearance modulating effect of the DOTA chelate and to develop PEG$_{11}$-DOTA conjugated 3-methyl-6-trimethylene-tetrazine **3** as an ADC activator (Fig. 2). To match the chemical properties of $^{177}$Lu- or $^{111}$In-labelled **4** and to avoid an osmolaric shock in mice treated with higher doses of activator, the DOTA chelate in **3** was complexed with nonradioactive lutetium(III) (we chose to use LuCl$_3$ as it has a 26-fold higher LD$_{50}$ than InCl$_3$). Next, we assessed the blood clearance and biodistribution of **3** spiked with $^{177}$Lu and found a clearance half-life of 12 min (Fig. 4a), very similar to the value for **4** and exceeding the value for 10 kD dextran derivatives[28,35]. Furthermore, [$^{177}$Lu]Lu-**3** had a complete lack of retention in tissues except some for the kidney, the organ of excretion (Fig. 4b and Supplementary Table 6).

**In vitro release and cytotoxicity.** Importantly, the new activator **3** gave 90% drug release upon reaction with **tc-ADC** in PBS for 1 h at 37 °C, as evidenced by protein mass spectrometry showing efficient conversion of the ADC to the fully activated product bearing four pyridazine conjugates derived from **3** without MMAE (31719.60 Da for the diabody monomer; Figs. 1 and 5a, and Supplementary Figs. 2a, b and 4). This yield corresponds well with the 80–85% yield found for the small 3,6-dimethyl-tetrazine in previous studies[19,21,23,32]. In serum, the liberation of MMAE from **tc-ADC** by **3** occurred at a slightly slower pace, giving 51% at 1 h and a maximal release of 80% at 20 h (Fig. 5b, c). The actual

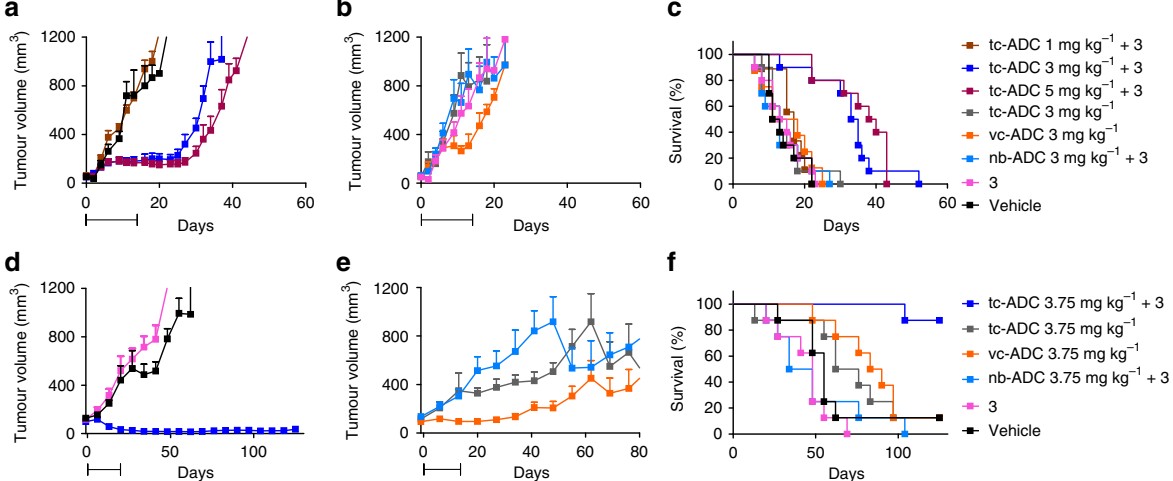

**Fig. 6** Therapeutic efficacy of ADCs in **a–c** LS174T colon carcinoma and **d–f** OVCAR-3 ovarian carcinoma mouse models ($n = 7$–10). Mean LS174T tumour volumes (with s.e.m.) in mice that within 2 weeks received i.v. **a** Four cycles of the combination of **tc-ADC** (1, 3, 5 mg kg$^{-1}$) with activator **3** (48 h post-ADC; 0.335 mmol kg$^{-1}$) or four cycles of vehicle, in comparison to **b** mice that received i.v. four cycles of **tc-ADC** alone (3 mg kg$^{-1}$), **3** alone (0.335 mmol kg$^{-1}$), enzymatically cleavable **vc-ADC** (3 mg kg$^{-1}$) or the combination of non-binding control (**nb-ADC**, 3 mg kg$^{-1}$) with **3** (48 h post-ADC; 0.335 mmol kg$^{-1}$). Mean OVCAR-3 tumour volumes (with s.e.m.) in mice that within 2 weeks received i.v. **d** Four cycles of the combination of **tc-ADC** (3.75 mg kg$^{-1}$) with **3** (48 h post-ADC; 0.335 mmol kg$^{-1}$), **3** alone (0.335 mmol kg$^{-1}$), or vehicle, in comparison to mice that received i.v. four cycles of **e tc-ADC** alone (3.75 mg kg$^{-1}$), **vc-ADC** alone (3.75 mg kg$^{-1}$) or the combination of **nb-ADC** (3.75 mg kg$^{-1}$) with **3** (48 h post-ADC; 0.335 mmol kg$^{-1}$). Survival curves for **c** LS174T and **f** OVCAR-3 bearing mice ($P < 0.0001$ and $P < 0.0002$, respectively; Mantel-Cox test). The bars below the x-axis indicate the treatment periods: four cycles of therapy or controls within 2 weeks (see injection scheme in Supplementary Fig. 21). Note: in **d** the error bars in the blue line (**tc-ADC** 3.75 mg kg$^{-1}$ + **3**) are obscured by the experimental points

maximal release yield is likely in between 80–90% as the liberated MMAE is not completely stable in serum solution for long timeframes and since the 0.1% trifluoroacetic acid (TFA) used in the protein mass spectrometry can favour the pyridazine elimination. The efficacy of this ADC system was reflected in cytotoxicity assays in LS174T and OVCAR-3 cell cultures. As opposed to tumours and spheroids, TAG72 expression is very low in cells grown in monolayers[36–39] and, therefore, the ADC activation was performed in solution in cell culture. The activator alone, shown to be completely stable in serum at 37 °C for at least 3 h (Supplementary Fig. 10d), was not toxic while **tc-ADC** alone only exhibited a relatively moderate toxicity with EC$_{50}$ values of respectively 71 nM and 29 nM. However, when a fixed dose of 3 µM activator **3** was combined with the ADC, the cytotoxicity increased 1000-fold, affording EC$_{50}$ values of 185 and 35 pM, matching the toxicity of the parent drug MMAE (Fig. 5d, e and Supplementary Table 1).

**In vivo reaction and release studies**. These results led us to examine the on-tumour reaction between **tc-ADC** and activator **3** using an in vivo blocking experiment[23]. In this approach, the sequential injection of **tc-ADC** and the nonradioactive activator (**3**) 2 days later is followed by administration of the highly reactive [$^{177}$Lu]Lu-**4** to bind to TCO linker that has not already reacted with **3**, thereby providing a read out on the reaction between **tc-ADC** and **3**. LS147T-tumour-bearing mice were i.v. administered $^{125}$I-labelled **tc-ADC** (2 mg kg$^{-1}$; 0.033 µmol kg$^{-1}$) and nonradioactive activator **3** (0.335 mmol kg$^{-1}$) 48 h later, followed by [$^{177}$Lu]Lu-**4** (0.335 µmol kg$^{-1}$) at 50 h. Biodistribution at 53 h confirmed the high tumour uptake of the ADC (Supplementary Fig. 16a) and revealed that there was no specific tumour uptake of [$^{177}$Lu]Lu-**4** (Fig. 4c), the value being as low as the uptake of [$^{177}$Lu]Lu-**4** without pre-injection of **tc-ADC** and activator[23]. In contrast, the positive control, **tc-ADC** followed by [$^{177}$Lu]Lu-**4** at 50 h, led to an efficient IEDDA reaction and a high

tumour uptake of the probe. Similar results were obtained for the OVCAR-3 model (Fig. 4c). These data demonstrate that the small activator **3** leads to a quantitative reaction of the extracellular tumour-bound **tc-ADC**. Also, this on-tumour reaction remained quantitative when the dose of **tc-ADC** was increased from 2 to 5 and 10 mg kg$^{-1}$ (Supplementary Fig. 16b, c).

To learn whether the high tumour uptake of **tc-ADC** and its complete reaction with **3** resulted in the expected high drug levels in tumour vs. other tissues, LS174T-bearing mice were administered **tc-ADC** with or without **3** 48 h later, followed by biodistribution at 72 or 96 h post-ADC administration. Liver and tumour homogenates and plasma were extracted with methanol followed by MMAE quantification with mass spectrometry (Fig. 4d). The activation of tumour-bound **tc-ADC** indeed gave high and sustained MMAE tumour levels 24 and 48 h after injection of **3**, indicating that tumour washout of MMAE, if any, is minimal. Furthermore, the MMAE levels were at least a 100-fold lower in liver and plasma, and in tumours that only received the ADC and not **3**, underlining the very favourable biodistribution of the ADC, its stability and its tetrazine-dependent release.

The presence of the DOTA chelate in activator **3** presented the opportunity to directly image the reaction of **3** with the tumour-bound **tc-ADC** by $^{111}$In-labelling and single photon emission computed tomography/computed tomography (SPECT/CT). Despite the ca. 100-fold lower reactivity of **3** ($k_2$ of $54.7 \pm 2.2$ M$^{-1}$ s$^{-1}$ with **tc-ADC**, Supplementary Fig. 11) compared to bispyridyl-substituted tetrazines[19], the intravenously administered [$^{111}$In]In-**3** specifically targeted the tumour-bound conjugate in mice that were administered **tc-ADC** 48 h earlier (tumour-to-muscle ratio of 1.7–2.1, Fig. 4e, g and i), as confirmed in a biodistribution study (Supplementary Fig. 16d). Moreover, even at the low imaging dose of 19 nmol kg$^{-1}$, four orders of magnitude lower than used in the blocking experiment, [$^{111}$In]In-**3** efficiently distributed throughout the tumour. As expected, a similar dose of the more reactive probe [$^{111}$In]In-**4** produced a higher tumour contrast (4.3–6.8 tumour-to-muscle ratio, Fig. 4f,

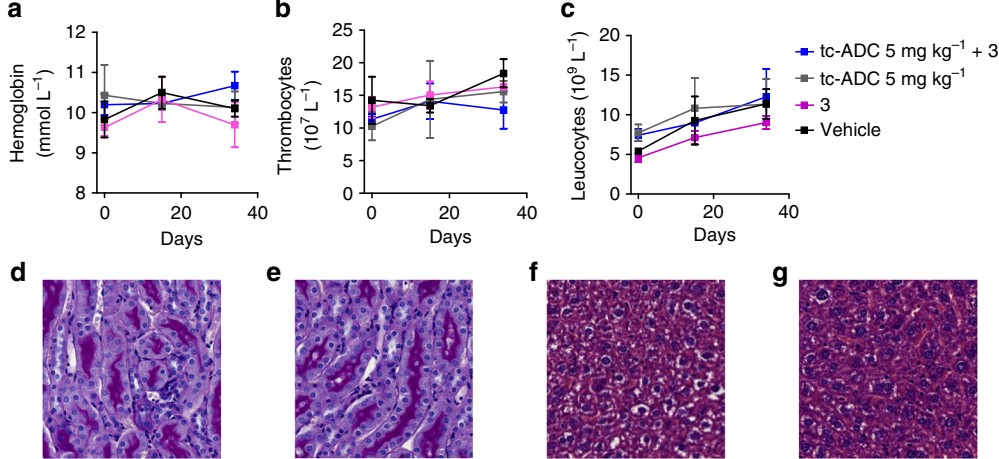

**Fig. 7** Haematologic toxicity and histopathology evaluation. Haematologic toxicity assessment showing similar **a** haemoglobin, **b** thrombocytes and **c** leucocytes values in tumour-free mice that within 2 weeks received i.v. four cycles of the combination of **tc-ADC** (5 mg kg$^{-1}$) and activator **3** (48 h post-ADC; 0.335 mmol kg$^{-1}$), the ADC alone, **3** alone or vehicle; data represent the mean with s.d. ($n = 3$). Histopathology: representative slices of renal cortex (**d**, **e**) and liver (**f**, **g**) obtained from tumour-free mice administered i.v. with four cycles of the combination of **tc-ADC** (5 mg kg$^{-1}$) and **3** (48 h post-ADC; 0.335 mmol kg$^{-1}$) (**d**, **f**) or vehicle (**e**, **g**)

h and i) shortly post-injection, suggesting its potential application as a companion diagnostic, to visualise ADC tumour uptake in patients before activation and/or to confirm drug release thereafter.

**Therapy studies.** On the basis of these favourable results we embarked on therapy studies in LS174T- and OVCAR-3-tumour-bearing mice. To evaluate the added benefit of the tetrazine-cleavable linker for non-internalising ADCs compared to the use of a protease-cleavable linker, an analogous TAG72-targeted diabody **vc-ADC** was prepared with the same valine-citrulline-linked MMAE as used in Adcetris (**2**, Fig. 2)[1]. For a non-binding control, maleimide-TCO-MMAE **1** was conjugated to an analogous diabody targeting the prostate specific membrane antigen (PSMA), affording **nb-ADC** (Fig. 2). As expected, while **vc-ADC** had the same uptake in LS174T and OVCAR-3 xenografts as **tc-ADC**, the non-binding control **nb-ADC** was not retained in the tumour (Fig. 3d). In addition, both ADCs exhibited the same low retention in blood and non-target tissues at 48 h post-injection as **tc-ADC**. The LS174T-tumoured mice received four cycles within 2 weeks of the combination of **tc-ADC** with **3** (48 h post-ADC; 0.335 mmol kg$^{-1}$). The mice that received 1 mg kg$^{-1}$ **tc-ADC** doses did not respond to therapy (Fig. 6a) as their tumours grew as fast as in the group that only received vehicle (17 days median survival, Fig. 6c, Supplementary Table 9). However, 3 and 5 mg kg$^{-1}$ **tc-ADC** doses gave a pronounced tumour growth delay of approx. 1 month ($P < 0.01$ at day 16 with respect to 1 mg kg$^{-1}$ dose and vehicle, one-way ANOVA with Bonferroni's post-test) and an extended median survival of 34 and 39 days. On the contrary, despite an initial transient effect of **vc-ADC**, this ADC and **tc-ADC** without activator **3**, the activator alone, and **nb-ADC** in combination with **3** all failed to control tumour growth when given in four cycles (Fig. 6b), leading to similar survival outcomes as found for the vehicle (12–14.5 days median survival; Fig. 6c). These findings indicate that **tc-ADC** is stable in vivo and that its therapeutic effect is dependent on TAG72-targeted tumour accumulation and the presence of **3**. Importantly, the protease-cleavable linker in the **vc-ADC** used here as a control was not effective in the extracellular matrix due to a lower and possibly slower MMAE release, most likely by extracellular proteases (Supplementary Fig. 17)[16].

A pilot study with OVCAR-3 tumour-bearing mice that received one single-dose of **tc-ADC** (0.75, 1.5, 3.75 or 7.5 mg kg$^{-1}$) followed by activator **3** (48 h post-ADC; 0.335 mmol kg$^{-1}$) already showed a dose-dependent therapeutic response, despite heterogeneous tumour growth (Supplementary Figs. 19 and 20). In a multi-dose study, the administration of four cycles of the combination of **tc-ADC** (3.75 mg kg$^{-1}$) with **3** resulted in a pronounced and durable tumour regression with barely palpable residual tumour masses until the end of the 4 month study, with 7 out of 8 mice surviving (Fig. 6d–f). On the contrary, most of the mice that received four cycles of the vehicle, activator **3** alone or **nb-ADC** in combination with **3** developed significantly larger tumours ($P < 0.05$ at day 20) and had to be removed from the study within 2 months (41–55 days median survival, Fig. 6d–f, Supplementary Table 8). Four cycles of **tc-ADC** alone or **vc-ADC** produced a heterogeneous response with a significantly larger mean tumour sizes in the second half of the study ($P < 0.01$ with **tc-ADC** alone at day 41; $P < 0.05$ with **vc-ADC** at day 83; Fig. 6e and Supplementary Fig. 22). While the limited therapeutic effect of **vc-ADC** is likely due to extracellular protease-based release[16], the minor effect of **tc-ADC** alone could suggest that some more non-specific MMAE release occurred at time points later than the earlier surveyed 96 h (for LS174T, Fig. 4d). Despite the partial therapeutic effect, these groups of mice had a limited median survival (72–86 days) and only one mouse per group reached the end of the study (Fig. 6f).

The click-to-release treatment was tolerated well, with no overt signs of toxicity that could be attributed to the use of the ADC, the activator or the combination. The LS174T xenograft is a very aggressive tumour model with a large health burden, as evidenced by the 5–10% weight-loss excluding tumour weight experienced in all groups, including the group that only received vehicle (Supplementary Fig. 23b). No weight losses were recorded in any of the groups of OVCAR-3 tumour-bearing mice (Supplementary Fig. 23a). Furthermore, the administration of four cycles of **tc-ADC**, activator **3** or **tc-ADC** followed by activator **3** had no effect on the levels of haemoglobin, leucocytes and thrombocytes in blood in tumour-free mice, and no alterations were observed in the histopathology of kidneys and liver, confirming lack of toxicity (Fig. 7 and Supplementary Fig. 24).

## Discussion

ADCs have the potential to improve chemotherapy of solid tumours, as they allow the use of toxins with orders of magnitude higher potencies. However, the current approaches are limited to efficiently internalising receptors, which are typically only over-expressed in a subset of solid tumour patient populations. There are non-internalising receptors and stroma proteins that would be attractive targets if there were a way to selectively release the drug extracellularly. We hypothesised that the use of exogenous chemistry instead of relying on endogenous drug release mechanisms may enable the development of a novel class of ADCs that does not require internalisation, complementing current approaches and thereby expanding the application scope of this powerful therapeutic modality. Herein we report the first example of therapy using bioorthogonally triggered drug release from a targeting agent in vivo. To this end, we developed a fast clearing ADC based on a diabody, designed to release MMAE through the IEDDA pyridazine elimination. While ADCs based on mAb fragments are usually hampered by low or modest target uptake due to fast clearance after i.v. injection, reducing the renal clearance rate by PEGylation led to a favourable combination of characteristics of mAbs and peptides. The diabody ADC matched the high tumour localisation of its parent mAb but cleared much faster from blood allowing the i.v. administration of the tetrazine activator 2 days later, leading to efficient extracellular activation and high MMAE tumour levels with no or minimal washout. This temporally controlled and traceless cleavage was independent from endogenous release mechanisms and tumour selective due to the very low retention of the diabody conjugate in non-target tissues, minimising toxicity. This is exemplified by the 2 orders of magnitude lower levels of MMAE in plasma and liver for **tc-ADC + 3**, and in tumour for **tc-ADC** alone. The lack of an Fc region and the resulting low systemic exposure may in the future also prove beneficial in limiting the off-target payload deposition that is currently hampering the ADCs in clinical trials[2]. In the course of developing the approach we found that the DOTA chelate can be used to reduce the clearance rate of certain tetrazine structures, thereby boosting in vivo reaction yields of tetrazine moieties that give high release yields. Furthermore, the use of a DOTA chelate enabled direct imaging of the reaction of the tetrazine with the tumour-bound ADC, demonstrating good intra-tumour distribution of the activator and potentially allowing future theranostic approaches. While the tetrazine dose was relatively high, the human equivalent dose (36 mg kg$^{-1}$) is, for example, still one to two orders of magnitude lower than the doses typically used for iodinated CT contrast agents in the clinic. The combination of ADC and activator led to a strong anticancer activity in vitro (EC$_{50}$ 185 and 35 pM), and in vivo in mice bearing, respectively, colorectal (LS174T) and ovarian (OVCAR-3) cancer xenografts. In addition to controlling tumour growth in the aggressive LS174T model, the click-cleavable conjugate resulted in a strong and durable response in OVCAR-3 mice at practical dose levels, leading to survival without any signs of toxicity for 4 months. On the contrary, the gold standard control, the same diabody conjugated with the protease-cleavable linker-payload combination used in the FDA approved Adcetris, was not effective in these tumour models, underlining the need for alternative activation mechanisms to complement the existing ADC therapies. Although a two-step approach requires the clinical development of two components and the optimisation of their dosing, this may partially be off-set by developing an ADC against a pan-carcinoma target and by the fact that the same activator can be used for different targets. In this respect, the CC49 diabody conjugate is very interesting as TAG72 is expressed in a wide range of solid cancers, but the click-to-release concept can also be extended to other targets and targeting agents.

Furthermore, extracellular cleavage may hold promise for a more homogeneous drug distribution in tumours, possibly even reaching cancer stem cells[8]. Another tempting avenue to explore is whether extracellular release can augment the recently observed activation of tumour-resident dendritic cells by tubulin inhibitors leading to antitumour immune responses[40]. In summary, with one FDA approved solid tumour ADC and approx. 5 more in pivotal trials there is a need to increase the number of viable targets. This study clearly suggests that a click-to-release system can expand the scope of the successful ADC concept to non-internalising receptors and stroma targets that have so far remained out of reach of ADC therapy.

## Methods

Additional methods and corresponding figures are provided in the Supplementary Methods and Supplementary Figures, including synthesis and characterisation of all new compounds (Supplementary Figs. 25–76) and diabody conjugates (Supplementary Figs. 1–3 and 8), radiolabelling (Supplementary Figs. 8 and 10a–c) and several in vivo experiments (Supplementary Figs. 14–24).

**Stability and activation of diabody conjugates in PBS.** The stability of stock solutions of diabody conjugates at 4 °C was monitored by quadrupole time-of-flight mass spectrometry (QTOF-MS). An aliquot of the stock solution (10 μL 2 μg μL$^{-1}$ in phosphate buffer pH 6.8 containing 2 mM ethylenediamine-tetraacetic acid (EDTA-PB) and 5% dimethylsulfoxide (DMSO)) was diluted with PBS (90 μL), and analysed with high-performance liquid chromatography (HPLC)-QTOF-MS. This procedure was repeated over the course of several months (6 months for **tc-ADC** and **nb-ADC**, and 18 months for **vc-ADC**). No degradation of the diabody conjugates was observed in this time frame (Supplementary Figs. 2 and 3). Aliquots of stock solutions of **tc-ADC** and **nb-ADC** (10 μL 2 μg μL$^{-1}$ in 5% DMSO/EDTA-PB) were diluted with PBS (90 μL), mixed with activator **3** (5 μL 2.5 mM in PBS; 1.25 × 10$^{-8}$ mol) and incubated at 37 °C for 1 h. Subsequent HPLC-QTOF-MS analysis demonstrated the formation of free MMAE ($m/z = +718.51$ Da) and the diabody reaction products without MMAE (Supplementary Figs. 4 and 5).

**Stability of activator 3 in mouse serum.** Lutetium-177-labelled activator **3** (11 nmol, ca. 20 MBq) was incubated in 50% mouse serum in PBS (0.7 mL) at 37 °C ($n = 3$). At $t = 0$, 30 min, 1, 2 and 3 h, 100 μL aliquots of the mixture were taken, treated with 100 μL ice-cold acetonitrile, mixed and centrifuged for 5 min at 13,000 rpm. More than 90% of the radioactivity was recovered in the supernatant at all time points. The supernatant was filtered through a 0.22 μm filter, 2.5-fold diluted with PBS and analysed by radio-HPLC. Linear regression of the % intact activator (peak at 10.3 min in Supplementary Fig. 10b) over time afforded an extrapolated half-life of ca. 54 h (Supplementary Fig. 10d).

**Activation of diabody conjugates in mouse serum.** **tc-ADC** stock solution (2 μg μL$^{-1}$ in 5% DMSO/EDTA-PB) was tenfold diluted with PBS. Subsequently 50 μL of this solution was twofold diluted with mouse serum and activator **3** was added (6.5 μL, 5 mM in PBS) followed by incubation at 37 °C. After 10 min, 1 h and 20 h aliquots of the solution were taken and proteins were precipitated by adding two parts of ice-cold acetonitrile. After vortexing, 10 min standing at −20 °C and centrifugation (13,000 rpm, 5 min), the supernatants were separated from the protein pellets, diluted with five parts of PBS and analysed by HPLC-QTOF-MS (Supplementary Fig. 7). MMAE recovery was quantified using calibration curves in 50% mouse serum, which were processed in the same manner. Reactions were performed in triplicate.

**Cell culture and proliferation assays.** The cell lines used in this study were obtained from the American Type Culture Collection. The human colon cancer LS174T and HT-29 cell lines were cultured in RPMI-1640 medium supplemented with 2 mM glutamine and 10% heat inactivated fetal calf serum (FCS). The human ovary carcinoma NIH:OVCAR-3 cell line was cultured in RPMI-1640 medium supplemented with 1 mM sodium pyruvate, 10 mM HEPES, 2 mM glutamine, 10 μg mL$^{-1}$ bovine insulin and 20% FCS. Twenty-four hours prior to the experiment, the cells were plated in 96-well plates at a 5000 cells/well density. Activator **3** (52 mM in PBS containing 5% DMSO), **tc-ADC** (2.36 μg μL$^{-1}$ in EDTA-PB containing 5% DMSO) and MMAE (63 μM in DMSO) were serially diluted in pre-warmed culture medium immediately before the experiment and added to the wells (200 μL final volume per well; $n = 3$). The **tc-ADC** was either added alone or followed by 3 μM activator (5 eq. with respect to the highest TCO concentration). After 72 h incubation at 37 °C, cell proliferation was assessed by an MTT assay. The proliferation assay was performed in triplicate. EC$_{50}$ values (Supplementary Table 1) were derived from normalised cell growth curves generated with GraphPad Prism.

**Animal studies.** The animal studies were performed in accordance with the principles established by the revised Dutch Act on Animal Experimentation (1997) and were approved by the institutional Animal Welfare Committee of the Radboud University Nijmegen. To determine the appropriate group sizes for the different experiments, the sample size calculation tool from the University of Iowa was used (http://homepage.stat.uiowa.edu/~rlenth/Power/index.html). Female BALB/c nude mice (7–9-week-old, 18–22 g body weight; Charles River Laboratories and Janvier) were subcutaneously inoculated ca. $3 \times 10^6$ LS174T cells (in 100 μL complete culture medium), ca. $5 \times 10^6$ NIH:OVCAR-3 cells (in 100 μL 1:1 RPMI-1640: matrigel) or $5 \times 10^6$ HT-29 cells (in 100 μL complete culture medium) in the hind limb. Tumour size was determined by caliper measurements in three dimensions (tumour volume = ½ × $l$ × $w$ × $h$) once per week (OVCAR-3) or three times per week (LS174T) by a blinded biotechnician. Biodistribution and imaging studies started when the tumours reached 0.1–0.2 cm³. Animals were randomly allocated to treatments groups. At the end of the studies, the animals were euthanized, blood was obtained by cardiac puncture and organs and tissues of interest were harvested, blotted dry, weighed, added with 1 mL PBS and the sample radioactivity was measured in a shielded well-type γ-counter together with aliquots of the injected dose to calculate % ID g$^{-1}$ and % ID. Stomachs and intestines were not emptied before γ-counting.

**ADC blood kinetics and biodistribution.** Two groups of tumour-free mice ($n = 4$) were injected with $^{125}$I-labelled **tc-ADC** at 1 and 5 mg kg$^{-1}$ dose (ca. 0.4 MBq in 100 μL saline per mouse). The mice were serially bled via the vena saphena (ca. 20 μL samples) at various times between 5 min and 72 h post-injection. Four days post-injection the mice were euthanized, one last blood sample was obtained via cardiac puncture and selected organs and tissues were harvested for γ-counting (Supplementary Table 4). The blood values were fit to a two-exponential curve and the pharmacokinetics parameters were calculated (Supplementary Table 3).

**ADC tumour targeting.** Groups of mice bearing subcutaneous LS174T, OVCAR-3 or HT-29 xenografts ($n = 3$–4) were injected with $^{125}$I-labelled **tc-ADC**, **vc-ADC** or **nb-ADC** at 2 mg kg$^{-1}$ dose (0.35–0.40 MBq in 100 μL saline per mouse). All mice were euthanized 48 h post-ADC injection and blood, tumours and other organs were harvested for γ-counting.

**Activator blood kinetics and biodistribution.** The activator precursor **S4** (see Supplementary Methods) was radiolabelled with $^{177}$Lu at ca. 1 MBq nmol$^{-1}$ molar activity and the resulting product was used to spike the nonradioactive activator **3**. Two groups of four tumour-free mice were injected with [$^{177}$Lu]Lu-**3** (0.335 mmol kg$^{-1}$, ca. 1.5 MBq in 130 μL PBS containing 5% DMSO per mouse). One group was euthanized 1 h post-injection. The second group was serially bled at various times between 2 and 90 min post-injection and euthanized 24 h post-injection. Selected organs and tissues were harvested from all mice for γ-counting (Supplementary Table 6).

**In vivo reaction between ADC and activator: tumour blocking studies.** Two groups of mice ($n = 4$) bearing LS174T or OVCAR-3 xenografts were injected with $^{125}$I-labelled **tc-ADC** (2 mg kg$^{-1}$, ca. 0.35 MBq in 100 μL saline per mouse) followed 48 h later by activator **3** (0.335 mmol kg$^{-1}$, in 130 μL PBS containing 5% DMSO) and, after 1 h, by [$^{177}$Lu]Lu-**4** (0.335 μmol kg$^{-1}$, ca. 1.5 MBq in 100 μL saline per mouse). Two more groups of mice ($n = 4$ for LS174T and $n = 3$ for OVCAR-3) were injected with the same amount of $^{125}$I-labelled ADC followed only by probe **4** 49 h later. All mice were euthanized 3 h post-probe injection and the $^{125}$I/$^{177}$Lu uptake in tumours and other organs and tissues was measured by γ-counting with a dual-isotope protocol with cross contamination correction. To evaluate the effect of ADC dose on the reaction efficacy with **3** (0.335 mmol kg$^{-1}$), three more groups of mice bearing LS174T xenografts were injected with $^{125}$I-labelled **tc-ADC** at 1, 5 and 10 mg kg$^{-1}$ dose (0.35–0.40 MBq in 100 μL saline per mouse) and evaluated as outlined above.

**In vivo reaction between ADC and activator or probe: SPECT/CT imaging.** Two LS174T tumour-bearing mice pretreated with **tc-ADC** (2 mg kg$^{-1}$, 48 h before activator injection) and two non-pretreated mice were injected with $^{111}$In-labelled activator **3** (13–19 nmol kg$^{-1}$, 13–20 MBq in 100 μL saline containing 100 μg gentisic acid per mouse), derived from precursor **S4**. Two more **tc-ADC** pretreated mice were injected with $^{111}$In-labelled probe **4** (ca. 19 nmol kg$^{-1}$, ca. 24 MBq in 100 μL saline containing 100 μg gentisic acid per mouse). The mice that received [$^{111}$In]In-**3** were euthanized 3 h post-activator injection, bladders were voided of residual radioactive urine and the mice were imaged post-mortem. The mice that received [$^{111}$In]In-**4** were anaesthetised 2 h post-injection and were imaged under anaesthesia (ca. 45 min imaging). Representative SPECT/CT images of a non-pretreated mouse injected with [$^{111}$In]In-**3** are depicted in Supplementary Fig. 18.

**Therapy studies.** Therapy studies where performed when tumours were clearly palpable (55 ± 26 mm³ LS174T tumour size, 5–7 days post-cell inoculation; 115 ± 43 mm³ OVCAR-3 tumour size, 6–8 weeks post-cell inoculation). A pilot (single-dose) therapy study in mice bearing OVCAR-3 xenografts is reported in the

Supplementary Information (Supplementary Figs. 19 and 20, Supplementary Table 7). Multi-dose therapy studies were performed following the injection scheme in Supplementary Fig. 21. Two groups of OVCAR-3 tumour-bearing mice ($n = 8$) received four cycles of TCO-containing ADC (**tc-ADC** or **nb-ADC**) at a 3.75 mg kg$^{-1}$ dose (in 100 μL PBS containing 5% DMSO) and activator **3** (0.335 mmol kg$^{-1}$ in 130 μL PBS containing 5% DMSO). Two groups of mice ($n = 8$) were injected with four cycles of **vc-ADC** or **tc-ADC** at the same dose followed by vehicle and, finally, two more groups of mice ($n = 8$) received four cycles of either the activator or vehicle. Four groups of LS174T tumour-bearing mice ($n = 8$–10) were injected with four cycles of increasing doses of **tc-ADC** (1, 3 and 5 mg kg$^{-1}$ in 100 μL PBS containing 5% DMSO) or **nb-ADC** (3 mg kg$^{-1}$) and activator **3** (0.335 mmol kg$^{-1}$ in 130 μL PBS containing 5% DMSO). Two groups of mice ($n = 10$) were administered four cycles of 3 mg kg$^{-1}$ **vc-ADC** or **tc-ADC** followed by vehicle and two more groups of mice ($n = 10$) received four cycles of either the activator or vehicle only. In both experiments, the animals were randomly grouped, monitored daily by experienced biotechnicians, and were removed from the study in case of poor physical condition (e.g., discomfort, reduced motility), in case of excessive body weight loss (>20% with respect to baseline or >15% in two consecutive measurements) or when tumours reached a 1 cm³ size. If none of these conditions occurred the mice were maintained in the study for up to 2 months (LS174T tumour model) or 4 months (OVCAR-3 tumour model). At mouse euthanasia, selected organs (kidney, liver and spleen) were harvested and formaldehyde-fixed for histopathology.

**Histology and haematological toxicity.** Four separate groups of tumour-free mice ($n = 3$) received four cycles of **tc-ADC** (5 mg kg$^{-1}$) and activator **3** (0.335 nmol kg$^{-1}$), **tc-ADC** only, activator only or vehicle, according to the injection scheme in Supplementary Fig. 21. Blood samples were collected from these mice before treatment (day 0), after treatment (day 15) and at euthanasia (day 34) to assess haematological toxicity (haemoglobin, leucocytes and thrombocytes levels). Kidney sections from these mice and from random mice in the multi-dose therapy groups were stained with hematoxylin-eosin and periodic acid Schiff reagent (PAS). Liver slices were stained with hematoxylin-eosin. Renal damage was microscopically graded from 0 (no damage) to 4 (severe damage) by an experienced pathologist, according to Supplementary Table 10[12].

**Data analysis.** Curve fitting, linear regressions and area-under-the-curve calculations were performed with GraphPad Prism (v 5.01). Group comparisons were performed with one-way ANOVA (with Bonferroni's post-tests) or two-tailed $t$-tests. Survival curves were compared with the log-rank (Mantel-Cox) test. Differences between data sets were considered significant when $P < 0.05$.

**Data availability.** The data that support the findings reported herein are available on reasonable request from the corresponding author.

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

## Acknowledgements

We thank Holger Grüll (University of Cologne) for the helpful discussions and critical reading of the manuscript, Sandra Heskamp and Otto Boerman (Radboud University Medical Centre) for helpful discussions and Danny Gerrits, Cathelijne Frielink, Janneke Molkenboer-Kuenen, Bianca Lemmers-van de Weem, Kitty Lemmens-Hermans and the staff at PRIME (Radboud University Medical Centre) for technical assistance. We thank Tong Zhu (Levena Biopharma) and Tilman Läppchen (Bern University Hospital) for contributing to the syntheses. We gratefully acknowledge the support of NanoNextNL (The Netherlands, Grant 03C.3). This work was supported by the Office of the Assistant Secretary of Defense for Health Affairs, through the Breast Cancer Research Program under Award No. W81XWH-15-1-0692. Opinions, interpretations, conclusions and recommendations are those of the author and are not necessarily endorsed by the Department of Defense.

## Author contributions

M.S.R., R.R., P.J.H. conceived the project; R.R., M.S.R., A.H.A.M.v.O., P.J.H., R.M.V., H. M.J. designed the studies; R.M.V., A.K., W.t.H. performed the organic synthesis; R.R. developed the radiochemistry; J.W. prepared the diabody conjugates; R.M.V., H.J.W. performed the ADC release studies with mass spectrometry; R.R., A.H.A.M.v.O. performed the in vitro and in vivo experiments; R.R., M.S.R., R.M.V., H.J.W., A.H.A.M.v.O., E.J.S, H.M.J., P.J.H. analysed results and/or edited the manuscript; M.S.R., R.R. wrote the manuscript.

## Additional information

**Competing interests:** R.R., A.H.A.M.v.O. and M.S.R. are employed by and shareholder of Tagworks. J.W. is employed by Avipep and P.J.H. is shareholder of Avipep. Both companies have a direct financial interest in the subject matter discussed in this manuscript. The remaining authors declare no competing interests.

