## [Peer Review File · Nature Communications]

Reviewers' comments:

Reviewer #1 (Remarks to the Author):

The manuscript submitted by Rossin et al describes the development and in vivo validation of non-internalizing antibody-drug conjugates (ADC) with the capacity to target TAG72 (an extracellular protein that is overexpressed in a range of human adenocarcinomas) and release their therapeutic cargo via a chemical trigger. The reported click-to-release strategy (which is based on the reaction of tetrazines with a trans-cyclooctene (TCO) modified to enable drug release, as originally reported by the same group in 2013 - ref 19), has been successfully implemented on a functional ADC construct, thus being significantly more advanced than the one reported in the original proof-of-concept study. By using a relatively short mAb-TCO linking moiety, the authors demonstrated the capacity of the ADC to maintain immunoreactivity, minimize TCO isomerisation to unreactive cis-cyclooctene, and enable drug release upon reaction with tetrazines.

The manuscript is very well written and the conclusions are supported by the experimental data. This is a novel work that examines the combined use of biologics and bioorthogonal strategies to optimize the specific delivery and release of anticancer therapeutics to solid malignancies. I believe the quality and therapeutic potential of this study justify publication in Nature Communications after addressing the following minor improvements:

Page 4: The fifth sentence of the second paragraph refers to Fig 1. However, that figure shows the new approach (not the previous one reported in ref 19). It would be clearer if "(Fig 1)" is moved to the third paragraph.

Page 5: Last sentence mentions that Suppl Fig 9 should show the stability of "ADC in serum at 37C for 24h (Supplementary Fig.9)". However, the caption of Supp Fig 9 says 1h incubation. Is this a typo? If not, could the authors provide the stability data at 37C for 24h?

Page 7: Fig 3 shows tc-ADC is specifically retained in the tumor, presumably by TAG72 targeting. This remarkable result was also corroborated in cell culture with LS174T and OVCAR-3 cells. However, the performance of the same experiment on a negative cell control (cells that do not express TAG72) would be necessary to complete the study.

Reviewer #2 (Remarks to the Author):

Rossin and colleagues describe a two-step pretargeted ADC that is compatible with non-internalizing surface antigens. The approach uses a PEG modified diabody for relatively rapid tumor uptake and clearance combined with an activator molecule to release MMAE from the antibody. They demonstrate good selectivity in vitro and excellent antitumor activity in vivo. The manuscript is clear and very well written. This approach may expand the potential targets for ADCs, especially for solid tumors.

Minor points

1. Although the two step approach is scientifically attractive, it is not popular with clinicians. Some discussion of the limitations of a pretargeting approach would be suitable. Also, it seems that quite a large amount of activator 3 is administered (~400 mg/kg). What is the equivalent dose (in mg) that would need to administered to a typical patient?

2. The bioactivity assay in Supplementary Figure 13 used to measure ADC binding activity is very rough. I don't think one can claim complete retention of immunoreactivity. A more sensitive assay such as direct ELISA against LS174T cells or mucin should be used to calculate EC50 of the ADC versus unconjugated diabody.

3. The results state that no drug was released from the ADC over 24 h but only one hour data appears to be shown (Supp. Fig. 9) . The 24 hour data should be included.
4. What is the detection limit of the assay used to determine ADC stability (i.e. Supplementary Fig. 9)? What percentage of drug release could be detected?
5. In Fig. 3d, why is the concentration of the specific ADCs significantly higher than the nb-ADC in blood, liver, lung etc. I suspect specific ADCs are slowly released from tumor cells back into the circulation. This may be a limitation of surface ADCs? Some discussion of this or alternative explanations should be provided in the text.
6. PK data such as in Fig 3a and Fig 4a are often plotted on a semilog (Y-axis log, x-axis linear) graph. This might be a better way for us to see the serum concentrations of the ADC or activator at relevant later times.
7. Fig. 4b shows activator biodistribution at 24 h but a shorter time such as 1 h seems more appropriate given the very short half-life of this compound.
8. More explanation of the assay shown in Fig 4C would help the reader to appreciate the results. Also, the actual concentration of MMAE in tumors versus blood and perhaps the liver should be shown. Methods for this have been published (for example, Burke et al, Mol Cancer Ther; 16(1); 116–23, 2017). This is important data and much easier for the reader to interpret.
9. In vitro cytotoxicity data should also include control ADCs and cancer cells that do not express TAG72.
10. In sup fig 34 and fig 6e, tc-ADC alone produced some long-term tumor regression. How do the authors explain this result since no activator was present?
11. Supp. Fig 36 does show that there is (likely significant) weight loss in mice treated with tc-ADC plus activator as compared to tc-ADC alone or nb-ADC + 3. Thus, it is not appropriate to say no toxicity was observed.
12. In supp Table 1, the 95% CI for tc-ADC + 3 is 16-22 pm but the EC50 is 185. It looks like CI should be 160-220?
13. Any idea why OVCAR-3 cells are so much more sensitive to MMAE as compared to LS-174T cells?

We would like to thank the reviewers for the careful reading of the manuscript and their constructive remarks. The following contains a point-by-point response to the comments, wherein our answers are in blue. The changes in the manuscript and Supplementary Information are marked in yellow.

Sincerely,
Marc Robillard

Reviewers' comments:

Reviewer #1 (Remarks to the Author):

The manuscript submitted by Rossin et al describes the development and in vivo validation of non-internalizing antibody-drug conjugates (ADC) with the capacity to target TAG72 (an extracellular protein that is overexpressed in a range of human adenocarcinomas) and release their therapeutic cargo via a chemical trigger. The reported click-to-release strategy (which is based on the reaction of tetrazines with a trans-cyclooctene (TCO) modified to enable drug release, as originally reported by the same group in 2013 - ref 19), has been successfully implemented on a functional ADC construct, thus being significantly more advanced than the one reported in the original proof-of-concept study. By using a relatively short mAb-TCO linking moiety, the authors demonstrated the capacity of the ADC to maintain immunoreactivity, minimize TCO isomerisation to unreactive cis-cyclooctene, and enable drug release upon reaction with tetrazines.

The manuscript is very well written and the conclusions are supported by the experimental data. This is a novel work that examines the combined use of biologics and bioorthogonal strategies to optimize the specific delivery and release of anticancer therapeutics to solid malignancies. I believe the quality and therapeutic potential of this study justify publication in Nature Communications after addressing the following minor improvements:

We thank the reviewer for these positive comments.

Page 4: The fifth sentence of the second paragraph refers to Fig 1. However, that figure shows the new approach (not the previous one reported in ref 19). It would be clearer if "(Fig 1)" is moved to the third paragraph.

We agree that it would be better to move the reference to Figure 1 to the next paragraph; this has been changed.

Page 5: Last sentence mentions that Suppl Fig 9 should show the stability of "ADC in serum at 37C for 24h (Supplementary Fig.9)". However, the caption of Supp Fig 9 says 1h incubation. Is this a typo? If not, could the authors provide the stability data at 37C for 24h?

We thank the reviewer for pointing out this mismatch between the text and what we showed in Supp Fig 9 (now Supp Fig 10). We have replaced the 1 h data by the 24 h data in this Figure.

Page 7: Fig 3 shows tc-ADC is specifically retained in the tumor, presumably by TAG72 targeting. This remarkable result was also corroborated in cell culture with LS174T and OVCAR-3 cells. However, the performance of the same experiment on a negative cell control (cells that do not express TAG72) would be necessary to complete the study.

We have now evaluated the biodistribution of the tc-ADC in a negative control tumor model (HT-29) and added it to the SI (Supplementary Table 5) and describe it in the main text. The tc-ADC uptake in this tumor model (0.80 ± 0.25 %ID/g) was significantly lower than that in LS174T and OVCAR-3 (29.39 ± 2.98 and 6.18 ± 0.40 %ID/g, respectively) but slightly higher than that of the non-binding ADC (nb-ADC) in the same tumors (0.2-0.3 %ID/g). This is due to the fact that, albeit being TAG72 negative in vitro in monolayers, the HT-29 cells do express very low target levels when grown as xenografts in vivo (see new ref 38 in the paper: Schlom et al., Dis Colon Rectum 1994, 37 (Suppl), S100-S105;). We believe the foregoing further supports the TAG72 specificity of tc-ADC, corresponding to earlier studies on the CC49 antibody and derivatives (please see refs 23, 25-28 in the paper for some examples).

Regarding the experiments in cell culture, as already noted for HT-29 above, unfortunately also most TAG72-positive tumor cells when cultured as monolayers express low to undetectable levels of TAG72, contrary to what is observed in the majority of, for instance, breast and colon carcinoma biopsy samples (see new ref 37 in the paper: Schlom et al., Cancer Res 1985, 45, 833-840). The LS174T was among the few cell lines to show low yet still detectable TAG72 levels in culture (ca. 1% of the cells expressing the target) but the expression was greatly enhanced when this cell line was grown as xenografts in mice (see refs 37 and 38). A similar behavior was also observed for OVCAR-3 cells, which were found to be TAG72 negative in monolayers (see new ref 39 in the paper: Metcalf et al., Int J Gynecol Cancer 1997, 7, 355-363) but showed heterogeneous TAG72 expression when grown as intraperitoneal ascites in mice (see new ref 40 in the paper: Starling et al., Cancer Res 1991, 51, 2965-2972). Furthermore, Schlom c.s. demonstrated enhanced target expression in LS174T cell colonies (agar) and spheroids with respect to monolayers (ref 37). Based on the selective expression of TAG72 in 3D structures, it has been hypothesized that TAG72 plays a role in the cell-cell interactions needed to establish such structures.

This phenomenon hampers EC50 experiments wherein cell monolayers are first incubated with the ADC followed by binding, washing of unbound ADC and incubation with the activator, as the ADC does not bind the cells. We have instead performed ADC activation in solution in the presence of the cells to show that, once the prodrug is activated in the cell vicinity, cytotoxic MMAE activity is restored. We have added a line in the main text to make this more clear.

We have now also performed the same EC50 experiment on HT-29 cells (see Supplementary Table 1, Supplementary Figure 20, and the caption of Figure 5) and, as expected for activation taking place in solution, we saw the same cell-killing power that was observed in LS174T and OVCAR-3 cell cultures.

Reviewer #2 (Remarks to the Author):

Rossin and colleagues describe a two-step pretargeted ADC that is compatible with non-internalizing surface antigens. The approach uses a PEG modified diabody for relatively rapid tumor uptake and clearance combined with an activator molecule to release MMAE from the antibody. They demonstrate good selectivity in vitro and excellent antitumor activity in vivo. The manuscript is clear and very well written. This approach may expand the potential targets for ADCs, especially for solid tumors.

We thank the reviewer for these positive remarks.

Minor points

1. Although the two step approach is scientifically attractive, it is not popular with clinicians. Some discussion of the limitations of a pretargeting approach would be suitable. Also, it seems

that quite a large amount of activator 3 is administered (~400 mg/kg). What is the equivalent dose (in mg) that would need to administered to a typical patient?

We agree with the reviewer that up till now two-step approaches have not been popular with clinicians. We think this is in part due to the therapeutic modalities that these two-step approaches pertained to. In the case of pretargeted radioimmunotherapy, pretargeting adds extra layers of complexity to a treatment modality that is already hampered by a high level of complexity, with issues in logistics (shipment of radionuclide, timely preparation of radiolabeled molecule), safety (radioactive dose to technician and patient), and referrals (transferring patient from the oncologist to the nuclear medicine department). In addition, the two non-pretargeted radioimmunotherapy products on the market, Bexxar and Zevalin, were against lymphoma and although they gave better the results than CHOP-R (chemotherapy plus non-radioactive antibody) the difference may not have been big enough to off-set the extra complexity at the time (although thankfully we are seeing a revival of this approach in the recent years). Another example of a two-step approach that never made it in the clinic is antibody directed prodrug therapy (ADEPT), which was and is greatly hampered by the immunogenicity of the exogenous enzyme that is targeted to the tumor to locally activate prodrugs.

A two-step approach to ADCs is indeed inherently more complex than the current ADCs, and will require a substantial investment to clinically develop the two components. In addition, the dosing of the two components will have to be optimized. We expect that this extra complexity will be acceptable if the ADC can be developed against a pan-carcinoma target, which will allow the therapy of tumors for which there are currently few therapeutic options (like ovarian cancer). In this respect, we would like to point to CAR-T's which, despite their complexity and very high costs, are being approved and are of high interests to biopharma and clinicians.

We have added the following lines to the discussion:

“Although a two-step approach requires the clinical development of two components and the optimization of their dosing, this may be in part off-set by developing an ADC against a pan-carcinoma target and by the fact that the same activator can be used for different targets.”

Regarding the dose of the activator, indeed this is relatively high at 453 mg/kg. Based on the FDA guidance (www.fda.gov/downloads/drugs/guidances/ucm078932.pdf) the human equivalent dose would be 8 % of that: 36 mg/kg. However, this is still one to two orders of magnitude lower than the doses typically used for i.v. administered iodinated CT contrast agents in the clinic, such as iodopamidol or iohexol (with MWs of ca 850 Da vs ca 1350 for activator 3). A typical iodopamidol dose is 700 mg/kg and the maximum iohexol dose in adults is 175 gram. Nevertheless, it is of course preferable to reduce the tetrazine dose. Recently Chen and co-workers reported that the ca. 10-fold more reactive 3-pyrimidinyl-6-alkyl-tetrazine motif gives a high release as well (please see ref 32). Using this motif in activators in future therapy studies should allow at least a 10-fold reduction in activator dose. We have added the following text to the discussion:

“While the tetrazine dose was relatively high, the human equivalent dose (36 mg/kg) is still one to two orders of magnitude lower than the doses typically used for iodinated CT contrast agents in the clinic. Furthermore, the recent finding that the ca. 10-fold more reactive 3-pyrimidinyl-6-alkyl-tetrazine motif gives a high release as well should allow markedly lower doses in future studies ³²”

2. The bioactivity assay in Supplementary Figure 13 used to measure ADC binding activity is very rough. I don't think one can claim complete retention of immunoreactivity. A more sensitive

assay such as direct ELISA against LS174T cells or mucin should be used to calculate EC50 of the ADC versus unconjugated diabody.

We agree with the reviewer that the immunoreactivity assay reported in the original version of our manuscript (which was adapted from Lewis et al., *Bioconjug Chem* 2016, 17, 485-492) was relatively rough. The gold standard of mAb immunoreactivity determination is the cell binding assay developed by Lindmo et al. (*J Immunol Meth* 1984, 72, 77-89) (ref 11 in the SI), where the immunoreactive fraction of a mAb is calculated from the linear extrapolation of binding to infinite antigen excess. However, as noted earlier, cell binding experiments with anti-TAG72 constructs are hampered by the low to undetectable levels of target expressed by most established cell lines when they are grown in monolayers. Therefore, for the revised version of the manuscript we have conducted a Lindmo-like assay adapted from Ngai & Reilley (*Appl Radiat Isot* 1993, 44, 1193-1197) (ref 10 in the SI). In this assay (Supp Figure 13), we incubated a constant amount of 125I-labeled tc-ADC with increasing amounts of bovine submaxillary mucine (BSM, comprising the TAG72 antigen) in solution followed by size exclusion chromatography purification, fraction collection and gamma-counting. The immunoreactive fraction of tc-ADC (94%) was then determined from linear extrapolation of the plot of total radioactivity / bound (T/B) vs. the inverse of BSM concentration in solution. We also determined the immunoreactivity of the unconjugated diabody (AVP04-58) and found a slightly lower value (84%), most likely due to the presence of a small amount of aggregates in solution, which were not present in the tc-ADC batch due to post-functionalization purification.

3. The results state that no drug was released from the ADC over 24 h but only one hour data appears to be shown (Supp. Fig. 9) . The 24 hour data should be included.

We thank the reviewer for pointing out this mismatch between the text and what we showed in Supp Fig 9 (now Supp Fig 10). We have replaced the 1 h data by the 24 h data in this Figure.

4. What is the detection limit of the assay used to determine ADC stability (i.e. Supplementary Fig. 9)? What percentage of drug release could be detected?

The limit of detection for MMAE in this assay is $1 \times 10^{-4} \mu\text{g}/\mu\text{l}$. The concentration of bound MMAE in the assay is $0.0472 \mu\text{g}/\mu\text{l}$. Therefore, the minimum percentage of free MMAE that we could have been detected is 0.2 %. As the amount of free MMAE is below $1 \times 10^{-4} \mu\text{g}/\mu\text{l}$, the percentage of bound MMAE is > 99.8 %. We have added this to the text preceding Supplementary Fig. 10 in the Supporting Information.

5. In Fig. 3d, why is the concentration of the specific ADCs significantly higher than the nb-ADC in blood, liver, lung etc. I suspect specific ADCs are slowly released from tumor cells back into the circulation. This may be a limitation of surface ADCs? Some discussion of this or alternative explanations should be provided in the text.

The concentration of the nb-ADC in various tissues is indeed lower than for tc-ADC and vc-ADC, and this is most likely due to a faster blood clearance of nb-ADC. The blood clearance of tc-ADC in tumor-free mice (Figure 3a) shows a blood level at 48 h which is similar to the blood level for tc-ADC in tumor bearing mice at the same time point (Figure 3d). Therefore, the blood levels of tc-ADC, and very likely also of vc-ADC, in tumor bearing mice (Figure 3d) are not due to shedding or washout from the tumor but instead reflect their natural clearance profile. The faster clearance of the nb-ADC results in less retention in other tissues as well. We have added a line to caption of Figure 3 to clarify this.

6. PK data such as in Fig 3a and Fig 4a are often plotted on a semilog (Y-axis log, x-axis linear) graph. This might be a better way for us to see the serum concentrations of the ADC or activator at relevant later times.

We have changed Fig 3a and Fig 4a into semilog graphs.

7. Fig. 4b shows activator biodistribution at 24 h but a shorter time such as 1 h seems more appropriate given the very short half-life of this compound.

We had selected the 24 h time point to see how where the activator is retained in mice after blood clearance is complete but the biodistribution at 1 h is also informative for the reason given by the reviewer. We have now conducted a biodistribution study at 1 h post injection and replaced the 24 h data with this data in Figure 4b. Supplementary Table 6 now shows the biodistribution at 1 and 24 h.

8. More explanation of the assay shown in Fig 4C would help the reader to appreciate the results. Also, the actual concentration of MMAE in tumors versus blood and perhaps the liver should be shown. Methods for this have been published (for example, Burke et al, Mol Cancer Ther; 16(1); 116–23, 2017). This is important data and much easier for the reader to interpret.

We have added additional lines in the text to better explain the set-up of the blocking experiment (Fig 4C).

In addition, following the method of Burke et al. we conducted in vivo experiments to measure the free MMAE levels in tumor, blood and liver in LS174T-tumor bearing mice treated with tc-ADC with and without activator 3. Please see the new Figure 4d and the extra paragraph, which shows that the tumor uptake of tc-ADC and its activation resulted in high and sustained MMAE tumor levels 24h and 48h after activation, indicating that tumor washout, if any, is minimal. Furthermore, the MMAE levels were more than 100-fold lower in liver and blood and in tumors that only received the ADC and not 3, underlining the favourable biodistribution of the ADC, its stability and its tetrazine-dependent release.

We also measured the MMAE levels in tumor, liver and blood arising from the vc-ADC at one timepoint, 24h post ADC injection. The MMAE values in blood and liver are comparable to tc-ADC while the value in tumor is lower. Please see Supp Fig 29 and the extra line in the main text.

9. In vitro cytotoxicity data should also include control ADCs and cancer cells that do not express TAG72.

Unfortunately, most TAG72-positive tumor cells when cultured as monolayers express low to undetectable levels of TAG72, contrary to what is observed in the majority of, for instance, breast and colon carcinoma biopsy samples (see new ref 37 in the paper: Schlom et al., Cancer Res 1985, 45, 833-840). The LS174T was among the few cell lines to show low yet still detectable TAG72 levels in culture (ca. 1% of the cells expressing the target) but the expression was greatly enhanced when this cell line was grown as xenografts in mice (see 37, and the new ref 38: Schlom et al., Dis Colon Rectum 1994, 37 (Suppl), S100-S105). A similar behavior was also observed for OVCAR-3 cells, which were found to be TAG72 negative in monolayers (see the new ref 39 in the paper: Metcalf et al., Int J Gynecol Cancer 1997, 7, 355-363) but showed heterogeneous TAG72 expression when grown as intraperitoneal ascites in mice (see the new ref 40 in the paper: Starling et al., Cancer Res 1991, 51, 2965-2972). Furthermore, Schlom c.s. demonstrated enhanced target expression in LS174T cell colonies (agar) and spheroids with respect to monolayers (ref 37). Based on the selective expression of TAG72 in 3D structures, it

has been hypothesized that TAG72 plays a role in the cell-cell interactions needed to establish such structures. Also HT-29 cells, which are often used as negative control in TAG72-related studies and are therefore now included in our study, express very low target levels when grown in vivo (ref 38). Accordingly, in our new in vivo study in mice bearing HT-29 xenografts we observed a low yet specific tumor accumulation of tc-ADC (0.80 ± 0.25 %ID/g).

However, this phenomenon hampers EC50 experiments wherein cell monolayers are first incubated with the ADC followed by binding, washing of unbound ADC and incubation with the activator, as the ADC does not bind the cells. We have instead performed ADC activation in solution in the presence of the cells to show that, once the prodrug is activated in the cell vicinity, cytotoxic MMAE activity is restored. We have added a line in the main text to make this more clear.

We have now also performed the same EC50 experiment on HT-29 cells (see Supplementary Table 1, Supplementary Figure 20, and the caption of Figure 5) and, as expected for activation taking place in solution, we saw the same cell-killing power that was observed in LS174T and OVCAR-3 cell cultures. In view of the above we do not think that doing this experiment for the nb-ADC will add more information.

10. In sup fig 34 and fig 6e, tc-ADC alone produced some long-term tumor regression. How do the authors explain this result since no activator was present?

While the new Figure 4d shows that the amount of free MMAE in tumor for mice treated with tc-ADC without activator is low at 24 and 48 h post injection, it may be possible that some more MMAE gets released at later time points due to proteolysis giving a slight therapeutic effect in the MMAE sensitive OVCAR cells. As can be seen from Supp. Figs 32 and 33 also lower doses of ADC + activator give some therapeutic effect in OVCAR mice, while this is not the case in LS174T mice (see Fig 6a). This seems to correspond with the fact that tc-ADC alone has some therapeutic effect in OVCAR but not in LS174T. We have now addressed this in the main text.

11. Supp. Fig 36 does show that there is (likely significant) weight loss in mice treated with tc-ADC plus activator as compared to tc-ADC alone or nb-ADC + 3. Thus, it is not appropriate to say no toxicity was observed.

We agree with the reviewer that Supp. Fig 36 (now Supp. Fig 38) seems to indicate that there is more weight loss for the group treated with tc-ADC + 3 compared to the apparent weight stability for the groups treated with tc-ADC alone, activator alone, vehicle or nb-ADC + 3. However, the apparent weight stability of the control animals was largely due to the development of very large tumors contributing to the total weight of the mice. At day 20 their tumors were ca 0.8 gram (corresponding to ca 4 % of body weight) heavier than the tumors in the treatment groups (3 and 5 mg/kg groups). Similarly, the apparent weight recovery at later time points of these treated mice (tc-ADC + 3) was the effect of tumor growth. In general, all mice bearing this very aggressive tumor line experienced an approx. 10% weight loss (excluding tumor weight) during the study, most likely due to the tumor burden, but showed no signs of discomfort. We had included this clarification in the section above Supp Fig 36 (now above Supp Fig S8), but we have now also added a note in the caption of Supp Fig 38.

For the above-mentioned reasons, we had and still have the following sentence in the paper referring to body weight excluding tumor weight “The LS174T xenograft is a very aggressive tumour model with a large health burden, as evidenced by the 5-10 % weight-loss excluding tumour size experienced in all groups, including the group that only received vehicle (Supplementary Fig. 38)”

For review purposes, please see the exemplary plot below showing the mouse weights excluding the tumor weights for the groups “tc-ADC + 3”, “vc-ADC”, and “vehicle”, indicating that the mice experience similar weight loss.

12. In supp Table 1, the 95% CI for tc-ADC + 3 is 16-22 pm but the EC50 is 185. It looks like CI should be 160-220?

We thank the reviewer for pointing out this error. We have replaced 16-22 pM with the correct CI: 158-217 pM.

13. Any idea why OVCAR-3 cells are so much more sensitive to MMAE as compared to LS-174T cells?

While there are literature examples on MMAE-based therapy in both tumor models, we could not find literature on the difference in MMAE sensitivity between the two cell types. It is believed that ovarian cancer cell lines are generally very sensitive to MMAE (in *Antibody-Drug Conjugates and Immunotoxins*, eds Lewis Philips, 2013; chapter 13, page 228 “Case Study: An Antibody-Drug Conjugate Targeting MUC16 for Ovarian Cancer” by Douglas Leipold and William G. Mallet), but we could not find literature on how this relates to the sensitivity of colorectal cancer cell lines such as LS174T. We believe that in addition to the difference in in vitro sensitivity, the difference in tumor growth rate plays an important role. LS174T is very aggressive and grows rapidly while OVCAR-3 grows much slower. For comparison please see the vehicle-treated groups in Fig 6a and Fig 6e. We could imagine that the rapidly growing LS174T can more easily overwhelm the available MMAE compared to OVCAR-3.

REVIEWERS' COMMENTS:

Reviewer #1 (Remarks to the Author):

In the opinion of this reviewer, all the suggested improvements have been satisfactorily performed. The revised manuscript should be published as is.

Reviewer #2 (Remarks to the Author):

The authors have carefully addressed all issues and questions and the revised manuscript is acceptable for publication.